# Risk Factors Associated with Children Receiving Treatment at Emergency Dental Clinics: A Case-Control Study

**DOI:** 10.3390/ijerph20021188

**Published:** 2023-01-09

**Authors:** Heba Jafar Sabbagh, Nuha Hamdi Albeladi, Nadeen Zouhair Altabsh, Nada Othman Bamashmous

**Affiliations:** 1Pediatric Dentistry Department, Faculty of Dentistry, King Abdulaziz University, Jeddah 22254, Saudi Arabia; 2Faculty of Dentistry, King Abdulaziz University, Jeddah 22254, Saudi Arabia

**Keywords:** risk factors, dental health services, emergencies, behaviour, health services accessibility

## Abstract

(1) Background: The process of managing children at the emergency dental clinic (ER-C) is a difficult challenge. This matched case-control study assessed risk factors associated with children visiting the ER-C compared to visits at the regular dental clinic (RD-C). (2) Methods: The participants included 421 children aged three to 12 years who were recruited at the ER-C (cases) and RD-C (controls) at King Abdulaziz University Dental Hospital, with each group matched for gender and age. A data-collection form was developed and validated in both Arabic and English, containing the following four sections: DMFT/dmft index, Frankl’s behaviour rating scale, Dental Neglect Scale, and Dental Care Barriers questionnaire. (3) Results: The ER-C (vs RD-C) group showed significantly more uncooperative behaviour (*p* = 0.002), a higher total mean dental neglect score (*p* = 0.003), and a higher dental barrier score (*p* < 0.001). Binary regression analysis showed that those making their first visit (AOR: 2.65, *p* < 0.001) and with higher dental barriers (AOR: 1.121, *p* < 0.001) were statistically significantly more associated with ER-C visitation. (4) Conclusion: These findings suggest that children who visit the ER-C are more prone to dental care barriers, uncooperative behaviour, and dental neglect, thus highlighting the importance of encouraging and planning their attendance to ensure optimal dental care.

## 1. Introduction

It is often very challenging to manage children who are suffering from caries at the emergency dental clinic (ER-C), especially in cases of heightened anxiety, uncooperative behaviour, and severe symptoms such as pain. In the same context, treatment may be limited, incomplete, or compromised due to insufficient time and materials [1], which increases the potential for side effects that degrade oral health [2]. Furthermore, those visiting the ER-C might have typically reached a stage of severe caries and symptoms, often including substantial pain. For example, a previous study found that nearly one-third of children visiting the ER-C had suffered for more than 30 days prior [2]. Moreover, ER-C treatment is not as cost-effective as prevention programmes and comprehensive treatment received at the regular dental clinic (RD-C) [3]. Overall, patients who seek care at the ER-C do not receive the same thorough dental care that is provided at the RD-C [4].

The American Academy of Paediatric Dentistry (AAPD) strongly recommends establishing a dental home care routine for children in early life, before 12 months of age [5]. Each child should have this opportunity, and treatment at the ER-C should be avoided as much as possible.

Despite these concerns, little is known about the risk factors associated with children visiting the ER-C for dental care rather than receiving comprehensive treatment at the RD-C. In this regard, children mainly rely on their parents to access dental treatment [6]; thus, parental neglect is a risk factor. Parental neglect occurs when parents fail to meet their children’s basic needs. A systematic review conducted in 2021 and included 10 studies to get an over-view of prevalence of dental neglect among children and possible risk factors. They found that the prevalence of dental neglect in children ranged from 34% to 56%. They also reported that dental neglect is associated with negative out-comes such as severe dental decay and untreated pain where the prevalence of untreated decayed teeth in the included studies was ranging from 38.9% to 99% [7].

Simultaneously, family socioeconomic status (SES) and access to care play important roles in dental health, especially for children. Many studies have reported that dental care barriers, such as geographic accessibility, appointment availability, scheduling issues, and affordability have some of the most significant negative impacts on oral health, often influencing delays in care-seeking [2,7,8]. In Saudi Arabia, researchers have focused on similar barriers associated with SES, including transportation difficulties, late appointment scheduling, fear of dental treatment, lack of perceived needs and/or awareness, limited knowledge of the health care system, and financial care costs [9,10,11]. Recently, the COVID-19 pandemic has created additional factors that negatively impact oral health care for children, including those pertaining to accessibility, care barriers, and dental neglect [12,13].

To date, the existing literature lacks evidence on factors that increase the risk for patients seeking treatment at the ER-C instead of the RD-C. Considering the serious problems that children may experience, it is imperative to clarify potential associations between family SES; individual behaviour; decayed, missing, and filled teeth (DMFT/dmft) index; dental neglect; and dental care barriers—especially in the COVID-19 context. Thus, this study assessed children’s behaviour, DMFT/dmft, dental neglect, and dental-care barriers as risk factors for visiting the ER-C instead of the RD-C among a sample of children receiving care at King Abdulaziz University Dental Hospital (KAUDH).

The null hypothesis is that children’s behaviour, DMFT/dmft, dental neglect, and dental-care barriers are not risk factors for visiting the ER-C.

## 2. Materials and Methods

### 2.1. Participants

This matched case-control study was conducted among children aged 3 to 12 years who received treatment at KAUDH between November 2021 and February 2022. Approval was granted by the research ethics committee at King Abdulaziz University (Approval No. 336-11-21; approval date: 8 December 2021). We divided the participants into groups according to whether they sought treatment at the paediatric ER-C (hereafter, cases) or regularly visited the RD-C (hereafter, controls). For every case recruited, a random control at the same hospital with matched age and gender was invited to participate. The inclusion criteria were: (1) healthy children (ASA1) [14], (2) aged three to twelve years, and (3) visiting paediatric dental clinics and (4) came for dental treatment except dental trauma. The sample size was calculated according to Kvist et al. [15] at a power of 80%; the estimate was 400 children, including 200 cases and 200 controls.

### 2.2. Methodology

Written questionnaires were provided to the parents and children, comprising the following three main sections.

#### 2.2.1. Section One

General information, including sociodemographic data on the child and parents (age, gender, parental education, and family income grouped to “Low” if <7000, “Moderate” if 7000 to 12,000 and “High” if >12,000 Saudi Riyal).The Dental Neglect Scale, which is a validated questionnaire [16] consisting of seven items (my child maintains his/her dental care, my child receives needed dental care in the dental clinic, needs dental care: parent defers, needs dental care: child defers, brushes his/her teeth twice per day, controls between-meal snacking, considers dental health important). Total scores are calculated for overall dental neglect, ranging from 7 (no neglect) to 35 (high neglect);Dental Care Barriers, which is a validated questionnaire consisting of 10 items that are designed to assess relevant factors, including geographic accessibility, scheduling-related barriers, and appointment availability and accessibility [17]. Total scores are calculated to reflect the overall degree of barriers, ranging from 10 (no barriers) to 46 (great barriers).

The questionnaire’s content validity was analysed and evaluated in both Arabic and English by five experts, including paediatric dentists and public health professors, for its relevance and clarity. The content validity index (CVI) score was 0.96.

#### 2.2.2. Section Two

An intraoral examination was performed based on the DMFT/dmft scores for each participant’s primary and permanent teeth [18].

#### 2.2.3. Section Three

This section was on child behaviour, as evaluated during the assessment visit via Frankl’s behaviour rating scale. It provides four group classifications according to the child’s attitude during dental treatment [19]. The four behaviour categories range from ‘Rating 1: definitely negative’ to ‘Rating 4: definitely positive’, described as follows: Rating 1: Definitely negative (− −) Refusal of treatment, crying forcefully, fearful, or any other overt evidence of extreme negativism; Rating 2: Negative (−) Reluctant to accept treatment, uncooperative, some evidence of negative attitude, but not pronounced; Rating 3: Positive (+) Acceptance of treatment, at times cautious, willingness to comply with the dentist, at times with reservation, but patient follows the dentist’s directions cooperatively; and Rating 4: Definitely positive (+ +) Good rapport with the dentist, interested in the dental procedures, and laughing and enjoying the situation. For the analysis, participants’ behaviours were categorised as either uncooperative (Ratings 1 and 2) or cooperative (Ratings 3 and 4) (see Table 1).

An inter-rater reliability test was conducted on 10 children to extract the DMFT/dmft scores by two examiners; it yielded a Kappa score of 0.92, indicating near-perfect agreement. For the other questionnaire sections, consensus was reached through several meetings, and discussions regarding the questionnaire answers and behaviour rating criteria.

### 2.3. Statistical Analysis

Descriptive statistics are presented as frequencies and percentages for categorical variables and means and standard deviations (SD) for continuous variables. Comparisons were conducted using Chi-square tests with nominal variables, *t*-test for continuous variables, and independent samples Mann–Whitney U Test for nonparametric data. A binary regression analysis was conducted to test the associations between ER-C treatment (dependent variable) and sociodemographic factors, DMFT/dmft, behaviour at dental clinics, dental neglect, and dental barriers (independent factors). The significance level was set to 0.05.

## 3. Results

This study enrolled 421 children, including 207 and 214 in the ER-C and RD-C groups, respectively. The ER-C group had a mean DMFT/dmft score of 8.77 ± 3.552, while the RD-C group scored 8.23 ± 3.411. There were no significant intergroup differences in other sociodemographic factors (*p* > 0.05). When asked whether they had received dental treatment in the RD-C over the last two years, only 91 (44.0%) participants in the ER-C group answered ‘yes’, versus 178 (83.2%) in the RD-C group (significant difference at *p* < 0.001). In addition, a statistically significant number of participants from the ER-C group reported that they had previously been treated in the ER-C (*p* = 0.012) compared to those in the RD-C group, who reported being treated in the RD-C significantly more com-pared to the ER-C.

Further, more participants in the ER-C group were reported by the dentist as uncooperative (i.e., 73 [61.3%] versus 46 [38.7%] in the RD-C, *p* = 0.002). There were no significant intergroup differences in DMFT/dmft (*p* = 0.113). Table 1 presents detailed information on this.

Table 2 shows the mean dental neglect score for the ER-C compared to the RD-C. The dental neglect total mean score was significantly (*p* = 0.003) higher for the ER-C (16.21 ± 4.447) compared to the RD-C (14.93 ± 4.203). More parents of the ER-C group agreed that their children maintained their dental care (*p* = 0.003) and brushed their teeth twice/day (*p* < 0.001) compared to those in the RD-C group. Meanwhile, more parents of the RD-C group considered dental health as important (*p* = 0.003) compared to those in the ER-C group.

Table 3 shows the mean dental barrier score for children visiting the ER-C compared to the RD-C. The dental barrier total mean score was significantly (*p* < 0.001) higher for the ER-C (18.89 ± 4.60) group compared to the RD-C (16.20 ± 3.509) group.

Table 4 shows a binary regression analysis of the association between visiting the ER-C and sociodemographic factors, previous dental visits, dental behaviour, DMFT/dmft, dental neglect, and dental barriers. Dental barriers (AOR: 1.121, 95% CI: 1.063 to 1.083, *p* < 0.001) and lack of previous dental visits (AOR: 4.864, 95% CI: 2.400 to 9.853, *p* < 0.001) were statistically significantly associated with increased AOR when visiting the ER-C. However, children who were younger showed a significant association with decreased AOR when visiting the ER-C (AOR: 0.329, 95% CI: 0.145 to 0.742, *p* = 0.007).

## 4. Discussion

This study investigated risk factors associated with children visiting the ER-C as opposed to the RD-C at KAUDH; it specifically assessed child behaviour, DMFT/dmft status, and markers for child dental neglect and dental care. In sum, the results revealed the prevalence of a higher number of perceived dental care barriers among children visiting the ER-C.

There were no significant differences in the mean DMFT/dmft scores between the ER-C and RD-C groups, indicating a high prevalence and severity of disease burden among the population. These results are consistent with reports from Al Agili et al. (2013), who found a high national prevalence of dental caries among children in Saudi Arabia (80% for primary dentition and 70% for permanent dentition) [20].

This study found that children visiting the ER-C are more likely to exhibit uncooperative dental behaviour compared to those visiting the RD-C. This finding is important as can help dentists to understand what to expect when treating children in the ER-C. It also put into consideration the importance of behaviour guidance and the use of different behaviour management techniques when treating children in the ER-C such as audio-visual distraction method, which was found to be more effective when compared with the tell, show and do method for modifying a child’s behaviour and decreasing anxiety specially among uncooperative children as reported by Thosar et al. [21]. Next, the ER-C group reported a significantly higher mean dental neglect score compared to the RD-C group, potentially due to a relative lack of oral health awareness. According to Sarkar et al. (2015), the Dental Neglect Scale measures how well a person cares for their teeth, receives professional dental treatment, and believes that oral health is essential [15]. Parents of children in the ER-C group considered dental health as more significantly important compared to those in the RD-C group; however, they reported more frequently that their children do not maintain their dental care, receive less than the required dental care at dental clinics, and infrequently brushed their teeth twice per day. Moreover, a significantly higher number of parents from the ER-C group reported that this was their child’s first dental visit compared to those in the RD-C group. This indicates that children treated in the ER-C do not receive dental care at home and do not access early prevention and treatment programs as recommended by the AAPD [5]. Additionally, preventive measures should be taken into consideration with ER-C group, such as community water fluoridation, fluoride toothpaste, fluoride varnish, and pit and fissure sealant as recommended by AAPD [22]. Moreover, modified restorative materials could be used with the ER-C group for instance: modified glass ionomer restoration, which has better mechanical and thermal properties, which is much suitable specially for uncooperative children as suggested by Chieruzzi M et al. (2018) [23].

Finally, the ER-C group reported a significantly higher Dental Care Barrier score compared to the RD-C group. In this context, the parents of the ER-C group reported that the greatest care barriers were related to appointment availability, accessibility, and scheduling. This supports the findings of Allaf et al. [17], who reported that, according to parents, schedule-related issues were the most common care barriers for their children. After adjusting the OR and overcoming the effects of confounding factors via the binary regression analysis, it was noted that participants in the ER-C group had primarily sought care in that location due to dental care barriers.

This study was subject to recall and desirability bias. As non-probability sampling was employed, the findings cannot be generalised to the whole population. However, the study sample was recruited from a university hospital that provides treatment to a heterogeneous population, including individuals of different nationalities. Moreover, the distributions of SES status and gender among the controls was similar to that in the general population, which addresses issues related to generalisation [24,25]. Another limitation of this study was the wide age range of participants, which could act as a confounding factor, thus affecting the outcome of the study. However, we tried to overcome this by matching the age of cases and controls. In addition, we conducted regression analysis, which helps to determine the unbiased relationship between two variables by regulating the effects of other factors. Unexpectedly, when matching confounders, older children showed an increased tendency to undergo ER-C treatment. This could highlight the prevalence of dental barriers, lack of dental home care routines, and absence of optimal dental care even in older children.

Furthermore, there was some overlap between the ER-C and RD-C groups regarding their previous dental visits to the ER-C or RD-C. However, this was addressed using the regression analysis, which statistically significantly indicated that almost five times more participants in the ER-C group ‘never’ visited dental clinics compared to those in the RD-C group.

These findings highlight the importance of planning for children who visit the ER-C to ensure the provision of adequate dental care, removal of dental care barriers, and improvement in the overall quality of oral health care. It also emphasises the need for directing parents and their children to the RD-C immediately after the ER-C treatment and establishing dental home care routines [5].

## 5. Conclusions

This study demonstrated that children visiting the ER-C are more prone to experiencing dental care barriers. Dentists should take into account the higher likelihood of uncooperative behaviour and dental neglect among such children. The findings emphasise the importance of encouraging and planning for such dental visits, with the aim of promoting and improving the awareness of optimal dental home care.

## Figures and Tables

**Table 1 ijerph-20-01188-t001:** Distribution of cases and controls according to their sociodemographic data, DMFT/dmft, and group (visiting emergency versus regular clinics); N = 421.

Variables		Dental Clinics Category		*p* Value	Total (%)
		Emergency N (%)	Regular N (%)		
Age (years)	<6	51 (24.6)	47 (22.0)		98 (23.3)
	6–9	96 (46.4)	98 (45.8)	0.706	194 (46.1)
	>9	60 (29.0)	69 (32.2)		129 (30.6)
Sex	Male	94 (45.4)	98 (45.8)	0.937	192 (45.6)
	Female	113 (54.6)	116 (54.2)		229 (54.4)
Maternal education	High school or less	119 (57.5)	123 (57.5)	0.998	242 (57.5)
	More than high school	88 (42.5)	91 (42.5)		179 (42.5)
Paternal education	High school or less	54 (26.1)	44 (20.69)	0.180	98 (23.3)
	More than high school	153 (73.9)	170 (79.4)		323 (76.7)
Family month income (SAR)	Low	59 (28.5)	48 (22.4)	0.182	107 (25.4)
	Moderate	102 (49.3)	104 (48.6)		206 (48.9)
	High	46 (22.2)	62 (29.0)		108 (25.7)
is this the child’s first dental visit	Yes	64 (30.9)	26 (12.1)	<0.001 *	90 (21.4)
	No	143 (69.1)	188 (87.9)		331 (78.6)
*-If yes*, did he visit dental emergency clinic?	Yes	83 (58.0)	83 (44.1)	0.012 *	166 (50.2)
	No	60 (42.0)	105 (55.9)		165 (49.8)
*-If yes,* was he previously treated in regular dental clinics?	Yes	88 (61.5)	174 (92.6)	<0.001 *	262 (79.2)
	No	55 (38.5)	14 (7.4)		69 (20.8)
The child lives with…	Single parent	23 (11.1)	23 (10.7)	0.905	46 (10.9)
	Both Parents	184 (88.9)	191 (89.3)		375 (89.1)
Visited dental emergency clinic in the last 2 years	Yes	83 (40.1)	83 (38.1)	0.562	171 (40.6)
	No	124 (59.9)	131 (61.2)		250 (59.4)
Child behaviour ‘Frankle classification’	Uncooperative (Rating 1&2)	73 (61.3)	46 (38.7)	0.002 *	119 (28.3)
	Cooperative (Rating 3&4)	134 (44.4)	168 (55.6)		302 (71.7)
DMFT/dmft	Mean ± SD	8.77 ± 3.552	8.23 ± 3.411	0.113	

* significance level at 0.05.

**Table 2 ijerph-20-01188-t002:** Mean dental neglect score for children visiting emergency clinics compared to regular dental clinics.

Variables	Dental Clinics Category		*p* Value	95% CI
	Emergency M ± SD	Regular M ± SD		
My child maintains his/her dental care	2.75 ± 1.274	2.38 ± 1.246	0.003 *	0.129 to 0.612
My child received needed dental care in the dental clinic	1.89 ± 1.039	1.69 ± 0.845	0.029 *	0.021 to 0.383
Needs dental care: parent defers	2.38 ± 1.374	2.24 ± 1.344	0.280	−0.117 to 0.404
Needs dental care: child defers	1.89 ± 1.208	1.73 ± 1.020	0.382 *^, w^	−0.059 to 0.369
Brushes her/his teeth twice per day	2.98 ± 1.166	2.37 ± 1.030	<0.001 *	0.396 to 0.818
Controls between-meal snacking	3.12 ± 1.122	2.85 ± 1.177	0.018 *	0.045 to 0.486
Considers dental health important	1.20 ± 0.535	1.69 ± 1.377	0.022 *^, w^	−0.695 to −0.292
Total neglect score	16.21 ± 4.447	14.93 ± 4.203	0.003 *	0.449 to 2.107

Likert scale range from ‘1 = strongly agree’, to ‘5 = strongly disagree’, ^w^ Independent-Samples Mann–Whitney U Test for nonparametric data, * significance level at 0.05.

**Table 3 ijerph-20-01188-t003:** Mean dental barrier score for children visiting emergency clinics compared to regular dental clinics.

Variables	Emergency M ± SD	Regular M ± SD	*p* Value
**Geographic accessibility**			
1. Do you travel for your appointments?	1.13 ± 0.338	1.11 ± 0.310	0.468 ^w^
2. How far is the specialized care centre from your residence?	2.80 ± 1.176	2.48 ± 1.141	0.004 *^, w^
3. How much time do you need to get to the specialized clinic?	2.53 ± 0.875	2.22 ± 0.881	<0.001 *^, w^
**Appointment availability and accessibility**			
4. What is the average waiting time for a dentist appointment?	2.37 ± 0.646	1.90 ± 0.642	<0.001 *^, w^
5. How long is the interval between two appointments?	2.19 ± 0.652	1.51 ± 0.603	<0.001 *
6. How easy/difficult is it to obtain school leave for your child’s dental appointment?	1.71 ± 0.977	1.89 ± 1.361	0.006 *
7. Can not get an appointment in two groups	1.76 ± 0.929	1.66 ± 0.930	0.272 ^w^
8. How many dental appointment have you missed because of your work?	2.14 ± 1.320	1.78 ± 1.046	0.002 *^, w^
**Scheduling related barriers**			
9. How many dental appointments have you missed because of your child’s school?	1.14 ± 0.471	1.11 ± 0.447	0.403 ^w^
10. How many appointments have you missed because you cannot fine caregiver for your children?	1.53 ± 0.929	1.24 ± 0.569	<0.001*^, w^
**Total Barrier score ***	18.89 ± 4.604	16.20 ± 3.509	<0.001 *^, w^

^w^ Independent-Samples Mann–Whitney U Test for nonparametric data; ‘Score 1 = less barrier’, to ‘5 = more barrier’, * significance level at 0.05.

**Table 4 ijerph-20-01188-t004:** Binary regression analysis of the association between children treated in emergency dental clinics and their sociodemographic factors, DMFT/dmft, dental neglect, and dental barriers.

Variables		AOR	95% CI	*p* Value
Maternal education	≤High school	0.709	0.446–1.128	0.147
	>High school	1		
Paternal education	≤High school	0.879	0.489–1.580	0.666
	>High school	1		
Family month income	Low	1.454	0.686–3.082	0.328
	Moderate	1.061	0.626–1.797	0.826
	High	1		
Child age	<6	0.329	0.145–0.742	0.007 *
	6–9	0.872	0.540–1.410	0.577
	>9	1		
Child gender	Male	0.921	0.607–1.396	0.698
	Female	1		
Child behaviour	Uncooperative	1.621	0.904–2.905	0.105
	Cooperative	1		
The child first visit	Yes	4.863	2.400–9.853	<0.001 *
	No	1		
The child lives with	Father or mother	0.786	0.433–1.427	0.429
	Parents	1		
DMFT/DMFT		1.009	0.943–1.079	0.805
Total neglect score		0.986	0.929–1.047	0.646
Total barrier score		1.121	1.063–1.183	<0.001 *

* significance level at 0.05.

## Data Availability

The data used in this study is available upon request from the corresponding author.

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
