# Peer review of "Risk Factors Associated with Children Receiving Treatment at Emergency Dental Clinics: A Case-Control Study"

_ijerph, 2023, doi:10.3390/ijerph20021188_

Round 1

Reviewer 1 Report

Interesting and generally well-structured work on oral differences in patients who refer to emergency structures compared to ordinary dental ones.

Only a few criticisms are present, listed below:

-check that all keywords are pubmed mesh terms

-at the end of the introduction section enter the null hypotheses of the study which will then be refuted at the end of the discussion section in the light of the results obtained in the study

- healthy patients are defined in the inclusion criteria. Better detail this aspect, both general health and dental terms. What were the parameters chosen in this regard?

- remove paragraph 2.2 which may be part of the previous one

- line 114 have the defined "experts" performed a preventive calibration procedure?

-line 122 the Frankl scale is a well known scale in scientific literature. Report the scores in a table and on in the text

- to make it easier to read the results, the significant p levels must be accompanied by an asterisk in the tables

-The discussion of the work appears too concise and substantially repetitive. First of all, some important considerations should be added regarding innovative treatment strategies in uncooperative patients, both in terms of prevention and materials. In this regard, I suggest to insert in the reference section the following scientific work that could be of help to the reader:

Chieruzzi, M.; Pagano, S.; Lombardo, G.; Marinucci, L.; Kenny, J.M.; Tower, L.; Cianetti, S. Effect of nanohydroxyapatite, antibiotic, and mucosal defensive agent on the mechanical and thermal properties of glass ionomer cements for special needs patients. J. Mater. 2018, 33, 638–649.

-Another extremely important aspect is represented by the possible effect that operative techniques could have on the biological characteristics of the oral cavity of these patients. In this regard, I request that the following scientific work be included in the discussion and in the reference section:

Carli E, Pasini M, Lardani L, Giuca G, Miceli M. Impact of self-ligating orthodontic brackets on dental biofilm and periodontal pathogens in adolescents. J Biol Regul Homeost Agents. 2021;35(3 Suppl. 1):107-115. doi:10.23812/21-3supp1-13

Ultimately, from the discussion section, before discussing the results of the study, I expect a complete and updated description of what, in terms of prevention and innovative therapy, can be included in these patients who, being uncooperative and with oral health levels worst represent a dental and general public health problem.

Reviewer 2 Report

This is a study comparing children who seek care in a regular clinic (RD-C) with those using an emergency clinic (ER-C). There are some areas to address that would greatly improve the report.

1. The age range is far too large and because of that, some grouped findings probably are not indicative of the respective groups. For example, behavior is likely to be different for very young kids.

2. It appears that these two groups are actually very similar in terms of using both ER and regular dental services - both groups crossed over. So I would ask if they are really distinct groups. The analyses suggest they are not.

3. Is the interrater reliability stated for all parameters or just behavior? 

4. The means and standard deviations are drawn out too far.

5. If the numbers are adequate, authors should divide the sample into under 5 years and over in the respective groups. The current data analysis and presentation make it difficult to determine if characteristics are affected by the age distributions.

6. The conclusions are not supported by the data.

Overall, it is hard to see any real differences between the groups as the data are presented and analyzed. The introduction leaves out many findings related to characteristics of those seeking care in other places outside the authors' country. The overlap of emergency and regular care within each group suggests that these are not distinct groups.

Reviewer 3 Report

Dear Authors, 

I share your view that both the topic of barriers to dental care, and its relationship to other factors, is important in the contemporary comprehensive approach to dental treatment. Your manuscript is very well prepared and is in huge part ready for being published. There are some minor things missing:

- eg asterisk in Table 1 (child behavior - 0.002)

Table 3 may be a little confusing. I would rephrase it in a way that would be more easily comprehensible/straightforward.

Kind regards

Reviewer 4 Report

Dear Authors

It was interesting to read the manuscript "Risk factors associated with children receiving treatment at 2 emergency dental clinics: A case-control study". I had a few suggestions highlighted in the attached file. Please go through it

Good Luck

Round 2

Reviewer 2 Report

The authors have addressed my concerns and their willingness to look at them is very refreshing!